# Construction of a Human Immune Library from Gallbladder Cancer Patients for the Single-Chain Fragment Variable (*scFv*) Antibody Selection against Claudin 18.2 via Phage Display

**DOI:** 10.3390/antib13010020

**Published:** 2024-03-12

**Authors:** Brian Effer, Daniel Ulloa, Camila Dappolonnio, Francisca Muñoz, Isabel Iturrieta-González, Loraine Cotes, Claudio Rojas, Pamela Leal

**Affiliations:** 1Center of Excellence in Translational Medicine (CEMT) and Scientific and Technological Bioresource Nucleus (BIOREN), Universidad de La Frontera, Temuco 4811230, Chile; isabel.iturrieta@ufrontera.cl; 2Carrera de Biotecnología, Facultad de Ciencias Agropecuarias y Medioambiente, Universidad de La Frontera, Temuco 4811230, Chile; d.ulloa09@ufromail.cl (D.U.); c.dappolonnio01@ufromail.cl (C.D.); f.munoz16@ufromail.cl (F.M.); 3Department of Preclinic Science, Medicine Faculty, Universidad de La Frontera, Temuco 4810296, Chile; 4Carrera de Ingeniería Pesquera, Facultad de Ingeniería, Universidad del Magdalena, Carrera 32 No. 2208 Sector San Pedro Alejandrino, Santa Marta 470001, Colombia; cotescastro@gmail.com; 5Programa de Doctorado en Ciencias Médicas, Universidad de La Frontera, Temuco 4811230, Chile; claudio.rojas@ufrontera.cl; 6Centro de Estudios Morfológicos y Quirúrgicos, Universidad de La Frontera, Temuco 4811230, Chile; 7Department of Agricultural Sciences and Natural Resources, Faculty of Agricultural and Forestry Science, Universidad de La Frontera, Temuco 4810296, Chile

**Keywords:** immune library, phage display, monoclonal antibody, gallbladder cancer, claudin 18.2

## Abstract

Gallbladder cancer (GBC) is a very aggressive malignant neoplasm of the biliary tract with a poor prognosis. There are no specific therapies for the treatment of GBC or early diagnosis tools; for this reason, the development of strategies and technologies that facilitate or allow an early diagnosis of GBC continues to be decisive. Phage display is a robust technique used for the production of monoclonal antibodies (mAbs) involving (1) the generation of gene libraries, (2) the screening and selection of isoforms related to an immobilized antigen, and (3) the in vitro maturation of the affinity of the antibody for the antigen. This research aimed to construct a human immune library from PBMCs of GBC patients and the isolation of *scFv-phage* clones with specificity against the larger extracellular loop belonging to claudin 18.2, which is an important biomarker overexpressed in GBC as well as gastric cancer. The immune-library-denominated GALLBLA1 was constructed from seven GBC patients and has a diversity of 6.12 × 10^10^ *pfu* mL^−1^. After three rounds of panning, we were able to identify clones with specificity against claudin 18.2. GALLBLA1 can contribute to the selection, isolation, and recombinant production of new human mAbs candidates for the treatment of gastrointestinal cancers.

## 1. Introduction

Gallbladder cancer (GBC) is a very aggressive malignant neoplasm of the biliary tract with a poor prognosis [1]. It has a high incidence in Latin America and Asia, with Chile being one of the countries with the highest mortality rates [2], which constitutes a public health problem that deserves attention.

GBC cancer is associated with risk factors such as chronic inflammation, gallstones, *Salmonella enterica*, *Helicobacter pylori* [3], obesity, and socioeconomic conditions, among others [4,5]. Regularly, the symptoms of this type of cancer are confused with biliary colic, preventing its prompt diagnosis [6]. Currently, there are no specific therapies for the treatment of GBC or early diagnosis tools, with surgical removal of the organ being the only preventive treatment [6,7], while for cases of advanced progression, there are few treatment options that have a poor prognosis [1]. Therefore, the development of strategies and technologies that facilitate or allow an early diagnosis of GBC continues to be decisive.

A useful, practical, and efficient way for the diagnosis and treatment of this and other types of diseases consists of the use of monoclonal antibodies (mAbs) due to their high specificity in recognizing and binding at their n-terminal ends to an antigen (protein targets and/or biomarkers) to then coordinate their inactivation and destruction through their c-terminal effector portion [8]. The therapeutic efficacy of a mAb in cancer treatment varies depending on the tumor antigen and the diverse mechanisms of action of the mAb, ranging from receptor blockade to immune system activation and disruption of oncogenic signaling [9]. Mechanisms such as blocking ligand binding, conjugated mAbs, depending on the antibody conjugation, can be either non-conjugated, blocking binding to evade undesired biological responses, or conjugated, harnessing the benefits provided by mAbs like vehicules and therapeutic agents such as cytotoxic molecules, in addition to mechanisms such as blocking the signaling pathway and depletion of the target via Fc interaction [9]. The most used strategy for generating these mAbs consists of using hybridomas [10]; however, this technique presents a tedious, delayed, and highly expensive procedure [11].

A plausible alternative that allows for faster production of mAbs is through in vitro-directed protein evolution procedures, which involve (1) the generation of native and immune gene libraries [12] or synthetic [13] that encode single-chain variable fragment (*scFv*) mAbs displayed through phage (phage display) [14]; (2) the process of screening and selection of isoforms related to an immobilized antigen (target proteins and/or previously identified biomarkers) [15]; and (3) the in vitro maturation of the affinity or specificity of the antibody, through the insertion of random mutations and selection of the isoform with improved affinity [16].

Phage display technology enables the expression and selection of high-affinity recombinant antibodies in laboratories in a controlled manner. These antibodies are fused with proteins from the phage coat and displayed on its surface, facilitating the isolation of *scFv* and *Fab* fragments targeting specific cellular markers [14]. Antibody libraries can be generated from pre-immune donors using V domains from the germ line or synthetic genes [12,13]. The multivalent presentation of antibodies is crucial for efficient selection. Moreover, antibody selection in whole cells has proven effective in isolating antibodies targeting challenging antigens, such as those associated with tumors, multi-pass membrane proteins like clusters of differentiation, or highly processed cell products [17].

The most important advantages of using phage display technology for the production of mAbs against GBC protein antigens are that the procedures are completely in vitro, avoiding animal immunization and, therefore, their use and manipulation [18]; its high flexibility, robustness, and speed to produce mAbs at low cost, and the possibility of expressing only the portion of the mAb that has affinity for the antigen, i.e., mAb in *scFv* format.

This, through the construction of immune libraries with specific diversity focused on GBC, isolating the “*V*” genes [19] from peripheral blood mononuclear cells (PBMCs) of patients with GBC, to then amplify the library in recombinant bacteriophages that will present in the proteins of its surface (i.e., protein pIII in bacteriophage M13) the diversity of functional *scFvs* in the library. The selection of *scFvs* with desirable affinity for the antigen is carried out using an iterative selection process [15].

These technologies have become the “Goodwill” of various biotechnology companies, which use their gene libraries as tools for the production of mAbs, such as *Creative Biolabs* (www.creative-biolabs.com; accessed on 3 January 2024), *SinoBiological* (www.sinobiological.com; accessed on 3 January 2024), and *ProteoGenix* (www.proteogenix.science; accessed on 3 January 2024). Currently, there are 14 mAbs produced via phage display and approved by the FDA (Food and Drugs Administration) and/or the EMA (European Medicines Agency) for the treatment of various diseases in humans, including several types of cancers [15], also against tumor markers in dogs [20], influenza virus [21], as well as coronaviruses in humans [22,23].

One emergent biomarker from several types of cancer, especially for GBC, is claudin 18 (UniprotKB—P56856). Also called CLDN18, it is a membrane protein (about 27.7 kDa) that belongs to the claudin family. It is composed of 261 aa distributed across two extracellular domains (28–80 aa and 144–174 aa), four transmembrane domains (7–27 aa, 81–101 aa, 123–143 aa, and 175–195 aa), and three intracellular domains (1–6 aa and 102–122 aa). Its main function is associated with maintaining tight junctions, which regulate the exchange of molecules between cells [24]. Claudins are predominantly found in gastric [25], pancreatic [26], and pulmonary [27] tissues. Claudin 18 has two isoforms: CLDN18.1, mainly expressed in the lung, and CLDN18.2, a specific isoform overexpressed in the stomach, which has emerged as an ideal biomarker [24] because it is widely expressed only in cancer cells, particularly gastric and gallbladder cancer [28].

Under normal conditions in gastric tissue, splice variant 2 of CLDN18 is located within tight junction complexes between gastric mucosal cells, and CLD18.2 epitopes have limited exposure to circulating antibodies. However, during the transition to a malignant state, loss of cellular polarity causes the CLDN 18.2 epitope to become exposed, making it more readily recognizable by antibodies [29]. The claudin 18.2 isoform represents a promising target for new experimental drugs such as zolbetuximab (IMAB362), currently under investigation in several clinical trials for advanced gastrointestinal tumors [29]. CLDN18.2 is retained in the presence of a malignant transformation, making it an ideal candidate for mAb bindings [30] via phage display to use in the GBC treatment and/or early diagnosis.

Herein, we describe the construction of an immune phage display library expressing *scFv* antibodies derived from PBMCs of gallbladder cancer patients and report the initial results of panning the library, where it was possible to isolate one clone with strong specificity against the large loop of the CLDN18.2 antigen.

## 2. Materials and Methods

### 2.1. Samples, RNA Extraction, and cDNA Synthesis

Seven PBMC samples from donors with GBC (case numbers: 552, 577, 716, 1125, 1288, 2199, and 2265) were acquired from Biobanco belonging to Pontificia Universidad Católica de Chile (Assignment 34) with prior approval by the ethics committee. Total RNA was isolated with a TRizol reagent (Invitrogen, Carlsbad, CA, USA), quantified using spectrophotometry, and stored at −80 °C. A total of 2 ug of RNA was used for the first strand of cDNA synthesis using the Affinity Script cDNA synthesis kit (Agilent, Santa Clara, CA, USA). The cDNAs were stored at −20 °C.

### 2.2. Amplification of Variable Genes (VH and VL) from Immunoglobulins

The synthesized cDNA was used as a template to amplify the PCR variable high (VH) and variable light (VL: *kappa* or *lambda*) regions from immunoglobulins, using primers described by Kugler et al. [31]. The PCR reaction contained a mix of four dNTPs at 10 mM each, 5 U of Taq polymerase (New England Biolabs, Ipswich, MA, USA), and 10 µM for each forward and reverse primers. The PCRs were subjected to 30 rounds of amplification at 95 °C for 1 min (Denaturation), 63, 58, and 61 °C (Annealing) for VH, *kappa* and *lambda*, respectively, for 1 min; and 72 °C (Extension) for 1 min. PCR reactions for VH, *kappa*, or *lambda* genes were resolved and isolated from 10 gL^−1^ agarose gels using the E.N.Z.A Gel Extraction kit (Omega Bio-Tek, Norcross, GA, USA). Three pools with the different fragments (VH, *kappa*, and *lambda*) of each patient were created, concentrated using a speed-vacuum concentrator (Eppendorf, Hamburgo, Germany), and quantified using a NanoDrop One (Thermo Fisher Scientific, Vacaville, CA, USA).

### 2.3. Cloning of VH and VL Segments into the Phagemid to Create the Library

A second PCR was performed for cloning VH and VL segments into the phagemid *pCDisplay3* (Creative Biogene, Shirley, NY, USA). Those primers were designed containing a *BssHII* and *SaIl* restriction site at the 5′ and 3′ ends for *kappa* and *lambda* segments, respectively. For VH segments, *XhoI* and *NheI* restriction sites were introduced at the 5′ and 3′ end, respectively. The PCRs were subjected to 30 rounds of amplification at 95 °C for 1 min (denaturation), 70, 74, and 75 °C (annealing) for VH, *kappa*, and *lambda*, respectively, for 1 min; and 72 °C (extension) for 1 min. The amplicons were resolved, isolated, and concentrated as described before, obtaining three pools: VH, *kappa*, and *lambda*.

We carried out the library assembly in two steps. In the first one, we digested kappa and lambda segments separately with *BssHII*/*SaIl* fast-digest enzymes (Thermo Fisher Scientific) for 30 min. The digested fragments were purified with 10 gL^−1^ agarose gel and ligated to 4 °C overnight with T4 DNA ligase (New England Biolabs, Ipswich, MA, USA) into a final volume of 20 µL with 500 µg *BssHII*/*SaIl* linearized *pCDisplay3* in a molar ratio 3:1 (VL:vector). The total volume of the ligation reaction was used to transform 200 µL of *XL1-Blue MRF′* competent cells (Agilent) and maintained at 4 °C for 10 min. The mixture was incubated at 42 °C for 1.5 min and returned to 4 °C for 2 min. The cells received 800 µL of SOC broth (Thermo Fisher Scientific, Vacaville, CA, USA) and were maintained at 37 °C, 150 rpm for 60 min. To determine the amount of transformants, 10 µL was taken and dilution series down to 10^−6^ were performed and plated out on 2xTY—GAT agar plates (Tryptone, 17 g L^−1^; yeast extract 10 g L^−1^; NaCl 5 g L^−1^; glucose 100 mM; ampicillin 100 µg mL^−1^; tetracycline 20 µg mL^−1^; and agar 15 g L^−1^) and incubated overnight at 37 °C. The remaining 990 µL was plated out on a 2xTY—GAT agar square dish (SPL Life Sciences, Pochon, Republic of Korea) and incubated overnight at 37 °C. All colonies were floated off on the pizza plate with 20 mL 2xTY, and the recombinant plasmids were extracted with the E.Z.N.A. As described before, the plasmid DNA Maxi kit (Omega Bio-Tek, Norcross, GA, USA) was finally used to concentrate and quantify two groups of recombinant plasmids: *pCDisplay3*_*kappa* and *pCDisplay3*_*lambda*.

In the second step, VH segments were digested with *XhoI/NheI* fast-digest enzymes (Thermo Fisher Scientific, Vacaville, CA, USA) for 30 min. The digested fragments were processed and cloned into the 500 µg *XhoI/NheI* linearized *pCDisplay3*_*kappa* and *pCDisplay3*_*lambda* separately, as described before. The total volume of the ligation reaction was used to transform 200 µL of *TG1* competent cells (Agilent, Santa Clara, CA, USA) and maintained at 4 °C for 10 min. The mixture was incubated at 42 °C for 1.5 min and returned to 4 °C for 2 min. The cells received 800 µL of SOC broth and were maintained at 37 °C, 150 rpm for 60 min. Seriated dilution was performed on 2xTY—GA agar plates to determine the number of transformants. The remaining 990 µL was plated out on a 2xTY—GA agar square dish and incubated overnight at 37 °C. All the colonies were floated off on the pizza plate with 20 mL 2xTY, and the recombinant plasmids were extracted with the E.Z.N.A. Plasmid DNA Maxi kit, concentrated and quantified like described before to finally have two groups of recombinant plasmids: *pCDisplay3*_*kappa_VH* and *pCDisplay3*_*lambda_VH*. Aliquots of each construction were mixed in equal parts and were subjected to a reaction with the Cre recombinase enzyme (New England Biolabs, Ipswich, MA, USA) following the manufacturer’s instructions to increase the diversity, as described by Sblattero and Bradburry [32]. The Cre recombinase reaction was used to transform 500 µL of *TG1* competent cells as described before. The size of the library was determined via serial dilution, taking 10 µL as described before. The remanent cells were plated out on a 2xTY—GA agar square dish and incubated overnight at 37 °C. All the colonies were floated off on the pizza plate with 20 mL 2xTY mixed with 4 mL glycerol. Aliquots of 1 mL were stored at −80 °C.

### 2.4. Library Packaging and scFv—Phage Production

To package the library, we used the methodology proposed by Ruschig et al. [12] modified as follows: we took one aliquot (1 mL) of the library to inoculate 50 mL 2xTY-GA and maintained to 37 °C to reach D.O._600nm_~0.5. Then, the culture was infected with 2.5 × 10^11^ colony forming units (*cfu*) with the helper phage *M13K07* (New England Biolabs) using a multiplicity of infection (MOI) 1:20 for 30 min without shaking and the following 30 min with 150 rpm at 37 °C. The culture was centrifugated for 10 min at 3200× *g*, and cells were resuspended in 250 mL 2xTY-AK (Kanamycin 20 µg mL^−1^) and maintained overnight at 150 rpm and 30 °C.

To recover the recombinant phages, cells were centrifugated at 10,000× *g*, and the supernatant was mixed with a PEG solution (PEG 8000 20% *w*/*v*; NaCl 2.5 M) in a proportion 1:5 (PEG solution:supernatant). The mixture was maintained at 4 °C for one hour, followed by centrifugation for one hour at 10,000× *g*. The supernatant was discarded, and phages were resuspended in a 1 mL phage dilution buffer (Tris-HCl 10 mM, pH 7.5; NaCl 2 mM; EDTA 2 mM) and maintained at 4 °C. A library aliquot was mixed with a protein loading buffer; then, it was denatured at 95 °C for 5 min and loaded onto a 12% and 20% SDS—PAGE. Protein observation was performed by staining the gel with Coomassie brilliant blue. For the Western blot analysis, proteins were transferred to a polyvinylidene fluoride (PVDF) membrane (Merck Millipore, Burlington, MA, USA) in a semi-dry electrotransfer system, using Owl^TM^ HEP (Thermo Fisher Scientific, Vacaville, CA, USA) for 60 min at 20 volts. A Western blot assay was performed using the Pierce^®^ Fast Western Blot kit, ECL substrate (Thermo Fisher Scientific, Vacaville, CA, USA), following the manufacturer’s instructions. Primary and secondary antibodies against his 6x (His-probe HIS.H8 sc-57598 and m-IgG_2b_BP-HRP—Santa Cruz Biotechnology, Dallas, TX, USA) and simius virus 5 (*SV5*) (V5-Probe C-9 sc-271944 and m-IgG_2a_BP-HRP—Santa Cruz Biotechnology, Dallas, TX, USA) tags (expressed in fusion with the *scFv* bound to the phages) were used.

### 2.5. Recombinant Production of Claudin 18

We used the expression vector *pET22b* containing an n-terminal *peIB* signal sequence for periplasmic localization, and it was acquired from Sigma-Aldrich. The nucleotide sequence of claudin 18.2 was retrieved from the *Kyoto Encyclopedia of Genes and Genomes* (KEGG) using the code 51208 and was used as a template to design primers forward (5′ggcggtctcccatgggccaccaccaccaccaccacgaaaacctgtattttcag 3′) including a n-terminal *6xHIS* and *TEV* (Tobacco etch virus) cleavage site; and reverse (5′ggcggtctccctcgagtcacactgcctgcagcatggc3′) against the first extracellular domain (cldn18.2), which was synthesized using GeneScript. The primer forward includes six histidine in the n terminal to facilitate protein purification by immobilized metal ion affinity chromatography (IMAC). The underlined sequences in the primers forward and reverse are the restriction sites of the *NcoI* and *XhoI* fast-digest enzymes (Thermo Fisher Scientific, Vacaville, CA, USA), respectively. The PCR was subjected to 30 rounds of amplification at 95 °C for 1 min (denaturation); 70 °C for 1 min (Annealing); and 72 °C for 1 min (extension). After the amplification, both the cldn18.2 and the plasmid *pET22b* were digested with *NcoI*/*XhoI* fast-digest enzymes. The ligation reaction between the gene and the plasmid was performed via the T4 DNA ligase (New England Biolabs, Ipswich, MA, USA), and the resulting plasmid (*pET22b_cldn18.2*) was transformed in *TOP10* competent cells (Thermo Fischer Scientific, Vacaville, CA, USA). Colony PCR was performed by taking *TOP10* clones as templates and using either primers for the insert (*cldn18.2*) or for the vector (forward: 5′ ccgcgaaattaatacgactcactatagg 3′; reverse: 5′gggttatgctagttattgctcagc 3′). Correct clones were confirmed using agarose gel electrophoresis (1%) based on the expected amplification size and corroborated by sequencing. The recombinant plasmid was isolated from *TOP10* cells using E.Z.N.A. Plasmid DNA Maxi kit (Omega Bio-Tek, Norcross, GA, USA). Overproduction and purification of cldn18.2 were performed in *E. coli* strain *Rosetta^TM^* (DE3) competent cells (Millipore, Burlington, MA, USA) as described by Cabarca et al. [33] modifying as follows: Luria–Bertani broth (LB) supplemented with ampicillin was inoculated with these cells harboring the recombinant plasmid and the cultures were incubated at 37 °C with shaking (150 rpm) until the A_600_ reached 0,6. Protein expression induction was completed with the addition of *Isopropyl 1-thio-D-galactopyranoside* (IPTG) at a final concentration of 1mM. The bacterial growth was continued in a shaker with 150 rpm for 3 h at 37 °C. Cells were harvested by centrifugation during 15 min at 7000 rpm at 4 °C. The cell pellets obtained during expression were resuspended in a lysis buffer (50 mM Tris-HCl, pH 8.0, 100 mM NaCl, 5% glycerol) containing DNase (5 µg mL^−1^), RNase (5 µg mL^−1^), lysozyme (1 mg mL^−1^), and 100 mM *Phenylmethanesulfonyl fluoride* (PMSF) (Sigma Aldrich, Burlington, MA, USA). Bacteria were disrupted via sonication (Fisherbrand^TM^ Sonicator 505—Fisher Scienctific, Pittsburgh, USA) on ice with 10 bursts of 20 s at amplitude 10. To obtain the supernatant, the lysate was centrifuged at 10,000× *g* at 4 °C for 30 min. The soluble extracts were applied with a syringe to a *HisTrap Excel* column 1 mL (GE Healthcare, Chicago, IL, USA) equilibrated with a binding buffer (50 mM Tris-HCl, pH 8.0, 100 mM NaCl, and 5% glycerol). The column was washed with five column volumes (CV) of a binding buffer containing Imidazole at 40 mM to remove the unbound material. The recombinant protein was eluted via the binding buffer with contained Imidazole at 250 mM. Imidazole was removed using a 3 kDa molecular weight cut-off Amico Ultra—four membranes (Merck Millipore, Burlington, MA, USA). The expression and purification of recombinant protein were monitored by SDS-PAGE electrophoresis and Western blot assays against 6xHis tag as was described above.

### 2.6. Selection of scFv—Phages through Panning

Library panning was performed as described by Barbas et al. [34], being modified as follows: 0.5 µg of recombinant cldn18.2, diluted in TBS (50 mM Tris-HCl, pH 7.5; 150 mM NaCl) was coated in a well of the 96-well ELISA plate (Corning Costar, Corning, NY, USA) during one hour at 37 °C. Then, the well was blocked using 3% (*w*/*v*) of BSA in TBS for one hour at 37 °C. For the first panning step, 10^10^ of the expanded library were incubated with the antigen coated in the well for two hours at 37 °C. Five washes were performed using TBS-T (0.1% Tween 20/PBS), and the elution of binding phages was performed with 10 mg mL^−1^ of Trypsin in TBS, and the round of panning was finished after superinfection of *TG1* cells by the phages retrieved in the trypsinization. After three panning steps, the eluted phages were mixed with TG1 cells for 30 min, and the cells were plated on LB agar-AK (ampicillin/kanamycin) and incubated overnight at 37 °C. Individual colonies were inoculated into 30 mL of 2xTY-AK at 150 rpm and 30 °C. The supernatant containing the *scFv* phage was stored at 4 °C for dot-blotting assays.

### 2.7. Phage Dot-Blotting Monoclonal Testing Antibody-Binding Specificity

Spots of the recombinant antigen cldn18.2 and BSA were coated on membranes of PVDF during 1 h at 37 °C. The membranes were washed three times with TBS-T and then blocked with 5% milk in TBS for 1 h at room temperature. Ten clones of *scFv* phages isolated in the third round of panning were added to the membranes and incubated for 2 h at 37 °C. Following five washes with a TBS-T buffer, the membranes were incubated for 1 h at 37 °C with a mouse-anti-*SV5* (1:1000) diluted in 5% (*w*/*v*) of fat-free milk in TBS. After five washes with the TBS-T buffer, secondary antibodies (m-IgG_2a_BP-HRP—Santa Cruz Biotechnology, Dallas, TX, USA) (1:5000) were added to the membranes and incubated for 1 h at 37 °C. Finally, the membranes were washed three times with TBS-T, and the substrate SuperSignal^TM^ West Femto Trial kit (Thermo Scientific, Vacaville, CA, USA) was applied to detect HRP activity via chemiluminescence using MyECL Imager (Thermo Scientific, Vacaville, CA, USA) equipment.

## 3. Results

### 3.1. Amplification of VH and VL Genes

RNAs of good quality were successfully purified and obtained from PBMCs belonging to seven patients with GBC. Their purity (A_260_/A_280_) oscillated between 1.7 and 2.4. All of them were used to synthesize cDNA, which was used to amplify VH and VL genes. Two groups of PCRs were performed. The first one was aimed at isolating the most putative and functional *V*-genes described in the VBASE (www2.mrc-lmb.cam.ac.uk/vbase/vbase-intro2.php accessed on 3 January 2024). The amplicons of “*V*” genes were around 300 pb (VH) and 500 pb (*kappa* and *lambda*) (Figure 1).

In the second one, primers, which contained some sites of restriction enzymes corresponding to the vector *pCDisplay3*, were used to clone the repertory isolated of twenty-eight *V* genes per patient (teen to VH, seven to *kappa*, and eleven to *lambda*). Overall, around 1064 PCR reactions were performed to have enough to digest and clone the *V* genes into the *pCDisplay3*. Finishing the PCR reactions, we created three pools of *V* genes (VH, *kappa*, and *lambda*), around 400 ng µL^−1^, with the restriction sites included in order to assemble the immune library.

### 3.2. Immune Library Assembly and scFv-Phage Production

To assemble the immune library, the VL genes (*kappa* and *lambda*) and *pCDisplay3* vector were double digested, using *BssHII* and *SaIl* enzymes (Figure 2), and ligated. After cloning and transforming XL1 blue cells, the pre-constructions reached 4.78 × 10^9^ (*pCDisplay3_kappa*) and 2.16 × 10^9^ (*pCDisplay3_lambda*). In a second step, those pre-constructions and VH genes were double digested, using *XhoI* and *NheI* enzymes (Figure 2) and ligated to form the pre-library: *pCDisplay3_kappa*_VH (3.5 × 10^8^) and *pCDisplay3_lambda*_VH (1.3 × 10^7^). Aliquots of these pre-library were mixtures in equal parts in a reaction with Cre recombinase enzyme to form the final library (GALLBLA1) inside TG1 cells, whose final size was 6.12 × 10^10^ *pfu* mL^−1^.

Once the GALLBLA1 infected with *M13K07* and obtained the *scFv*-phage particles, we monitored the library through Western blot analysis against the *SV5* tag, which is exclusive of the *scFv* phage particles produced. It was possible to observe a band around ~38 kDa (red arrow in Figure 3) positive to *SV5* tag fused to *scFv*.

### 3.3. Recombinant Production of the Antigen cldn18.2

In this research, we have truncated the larger extracellular loop (28–80 aa, Figure 4a), expressed in the prokaryotic system, because this loop does not have post-translational modification and because this system is cheaper than the eukaryote system. So, codons belonging to this loop were optimized for *E. coli* and subcloned into *pET22b* (Figure 4b). This loop is 6 kDa predicted via ProtParam [35], but including *6xHIS* and *TEV* tags, its weight must increase to 7.6 kDa. However, the SDS-PAGE and Western blot analysis showed a band around 15 kDa with high purity (Figure 4c,d).

### 3.4. Selection of scFv-Phages against cldn18.2

Panning through an ELISA plate was the methodology used in this study to select fragment antibodies from the GALLBLA1. After three rounds of panning, ten individual colonies of scFv-phages were isolated to test their specificity via cldn18.2 through dot-blotting, using the control protein used for the blocking (BSA) in panning to avoid false positives. As shown in Figure 5, clones 1 and 2 showed slight affinity by BSA (head red arrows) but not for cldn18.2; but clones 6, 7, and 10 (head blue arrows) recognized cldn18.2 without interaction with BSA; clone 6 being the candidate that has the most specificity to cldn18.2.

## 4. Discussion

Immune libraries play a very important role in the selection of antibodies against specific disease antigens, and they are based on the immune repertoire collected through *V* genes from individuals infected or suffering from a particular disease [36].

However, this kind of library is biased in producing mAbs, usually of high affinity against antigens associated with a specific disease [37], such as the constructed library reported here.

We used PBMCs from seven patients with GBC to isolate *V* genes. To maximize complementarity, we used degeneracy primers designed by Kugler et al. [31], which can isolate different representative families of antibody *V* genes.

All these families were cloned into *pCDisplay3* to ensemble the library. Our library size was compatible with the size reported to other constructed libraries, which range from 10^6^ to 10^10^ [38,39,40]. However, it is very laborious and time-consuming to build large libraries (>10^10^) due to both the use of a large number of blood samples [41] and bacteria transformations [42,43]. Shui et al. [39] built an immune library against liver cancer with blood samples from four patients, obtaining a library of size 1.7 × 10^7^. In the case of Wu et al., they used just three lymph node samples from colorectal cancer patients to build an immune library. This final library was 5.53 × 10^6^.

On the other hand, Dong et al. [40] built a naïve library with more than four hundred healthy people, using for the library assembly a LoxP peptide as a linker between the *V* genes to form *scFv*, in order to increase both the variability and the size of the library, using cre-recombinase enzyme. The final size of the library made by Dong et al. was 1 × 10^11^.

In our case, we use the same strategy previously proposed by Sblattero and Bradburry [32] for the construction of large libraries through the use of the Cre recombinase enzyme, using the LoxP site between the VH and VL in the phagemid *pCDisplay3* (Figure 2). However, regardless of the size obtained from the libraries, because of stop codons or deletions generated via PCR, frameshifts, and wrong immunoglobulin rearrangements [39], the real size of the functional library is always smaller than the one reported.

Different sizes of *scFv* produced have been reported through literature, and this will depend on the type of phagemid used in the library’s construction, the type of library, the expression system used, and the purification strategy of the *scFv* employed. Ghaderi et al. reported a 28 kDa *scFv* produced against an extracellular domain of checkpoint PD-1. They used a human semi-synthetic Tomlinson I+J phagemid libraries, and the monoclonal *scFv*-phage isolated was used to infect *E. coli* Rosetta-Gami 2 to the production of *scFv* [44].

On the other hand, Shams et al. isolated a *scFv* from a naïve human library against the larger loop from CD20. They recovered the sequence of *scFv* and cloned it into *BamHI/XhoI* restriction site *pET30a*. This vector adds several additional sequences into the *scFv*, such as two 6xHis tags (n and c terminal position), thrombin cleavage site, S-tag, and enterokinase site. Maybe, for this reason, they produced a 35 kDa *scFv* [45]. In our case, the *pCDisplay3* phagemid is able to codify two tags in frame with the *scFv* cloned 6xHIS and SV5 tag (2.3 kDa). For this reason, our *scFvs* pool obtained in the library amplification was around 38 kDa.

Tight junctions in cells are very important, and claudins have an important role in barrier, fence, intracellular signaling, ion interchange, migration, and proliferation [46,47,48]. In this sense, claudin 18.2 is highly expressed in tumors such as gastric cancer [49], pancreatic cancer [26], lung cancer [50], cholangiocarcinoma [51], and gallbladder cancer [28], and for this reason is one of the most clinically relevant targets in cancer immunotherapy [52].

Similarly, Shams et al. [45] expressed the larger loop from CD20 antigen in a prokaryote system to be used to isolate *scFv* antibody from a naïve library. CD20 resembles structurally cladn18.2, having two extracellular loops. In the same way as these results, the largest loops in the SDS-PAGE analysis showed a greater weight than expected [45]. In our cases, the recombinant loop we produced could be forming a dimer.

Currently, there are several strategies to isolate fragments of mAbs from gene libraries, such as antigen coated to magnetic beads [53,54], streptavidin-coated solid-phase extraction [55], directly on the cells [56], and microtiter plates [15]. For simplicity and to save time and resources, this research was applied to the pan using microtiter plates. After three rounds of panning, we isolated ten colonies in a random way to screening specificity by cldn18.2. Although a lot of colonies could be isolated in the last round of panning, most of them are unspecific for cldn18.2, but the identification of clone 6 has shown at least one *scFv* phage with high binding specificity by the claudin 18.2, one important antigen-related to gallbladder cancer as well as gastric cancer. Actually, there is just one chimeric mAb in review by the FDA and EMA called Zolbetuximab against claudin 18.2 to treat gastric or gastroesophageal junction adenocarcinoma [57]. An immune library like the one reported in this research can contribute to the selection, isolation, and recombinant production of new human mAb candidates for the treatment of gastrointestinal cancers. The future perspectives will focus on analyzing the sequences of the six CDRs of the selected clone 6, affinity maturation if applicable, recombinant production in different formats and its characterization; and in this way evaluate its usefulness for possible treatment or diagnosis.

## Figures and Tables

**Figure 1 antibodies-13-00020-f001:**
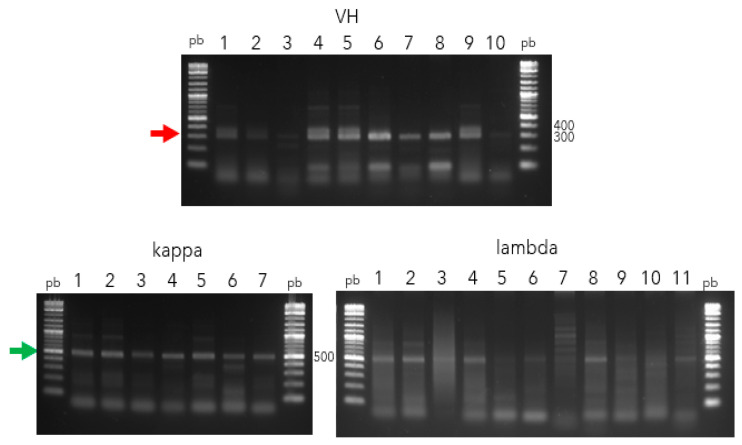
Amplification of *V* genes (VH, *kappa*, and *lambda*) from PBMCs belong to GBC patients. Lane pb, 1 kb Plus DNA ladder (New England Biolabs); Lane 1–10, 1–7, and 1–11 correspond to amplicons belonging to VH (red arrow), *kappa*, and *lambda* (green arrow) genes, respectively, resolved in 1% agarose gel.

**Figure 2 antibodies-13-00020-f002:**
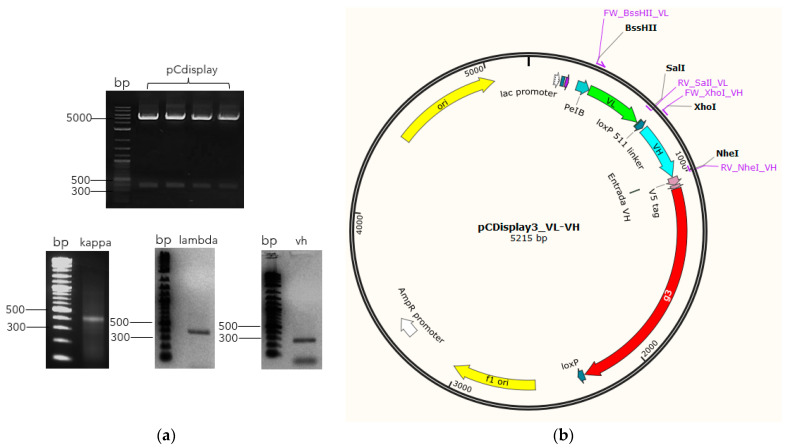
Library assembly schematic. Once the V genes have been amplified with their respective restriction sites added, they are digested in parallel with the vector (**a**) in order to be cloned into the restriction site corresponding to each group of *V* genes (*BssHII*/*SaIl* to *kappa* and *lambda* (green arrow), and *XhoI*/*NheI* to VH (blue arrow)) in frame with M13 phage minor coat protein g3 (red arrow) (**b**), to finally be transformed into bacteria cells.

**Figure 3 antibodies-13-00020-f003:**
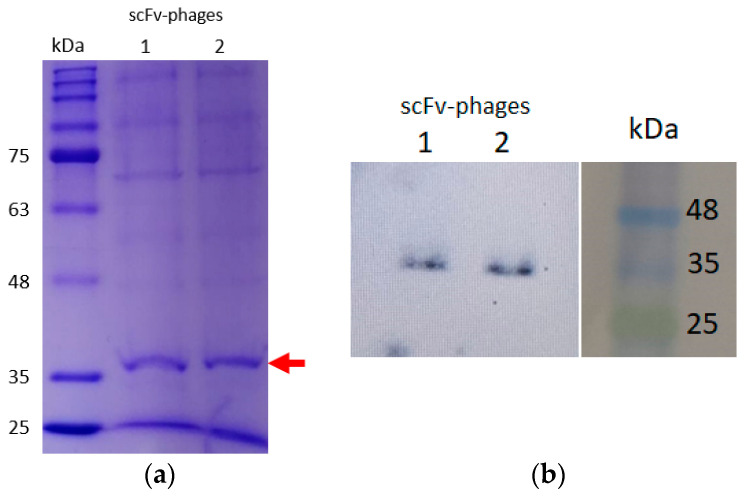
Resolution of the amplified immune library through infection with helper phage *M13K07*. A total of 12% SDS-PAGE of a pool of *scFv*-phage duplicated (**a**), identified with Western blot analysis. For best resolution, another 20% SDS-PAGE was performed and transferred into a PVDF membrane and hybridized with an anti-SV5 tag. The analysis identifies the same bands with ~38 kDa (red arrow) (**b**).

**Figure 4 antibodies-13-00020-f004:**
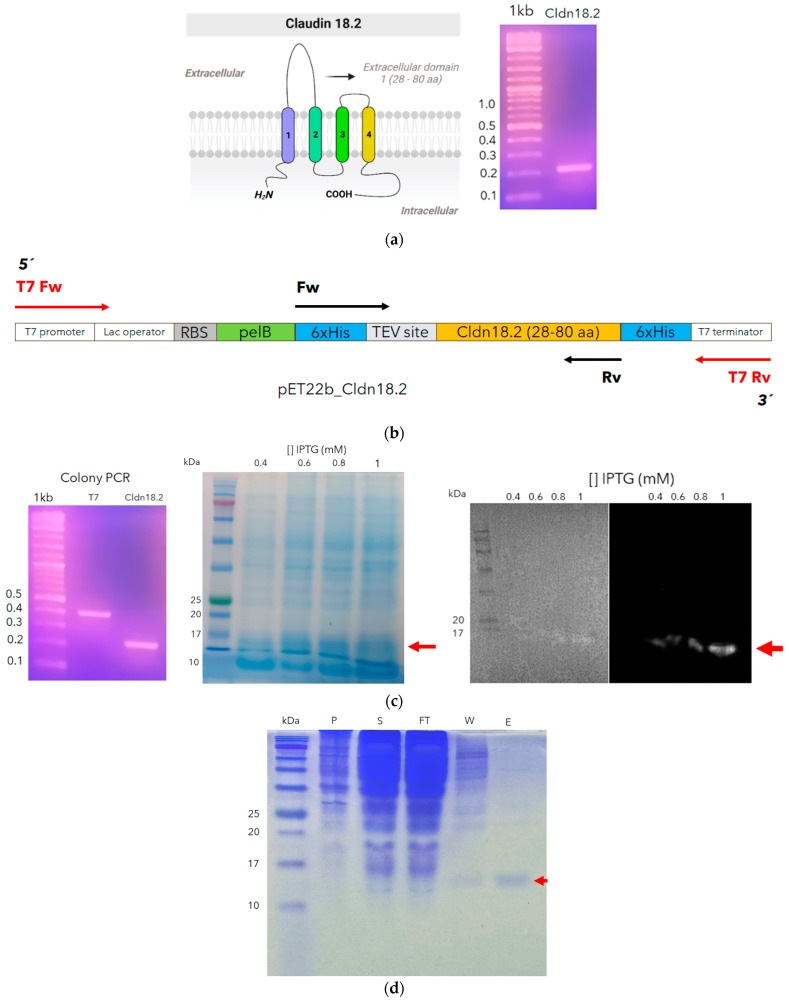
Cloning and recombinant expressing of an extracellular larger loop of claudin 18.2. The sequence encoding the larger loop (28–80 aa) was amplified. The cilindres with different colors, represent the four hydrophobic transmembrane domines of claudin 18.2) (**a**) and cloned into the pET22b in frame with 6xHis and TEV tag (**b**). Colony PCR was performed with T7 primers and those corresponding to the insert to confirm the right construction (**c**). An expression test was performed to identify the best concentration of ITPG; additionally, the protein expression was monitored through Western blot analysis (red arrows) (**c**). Finally, cldn18.2 was expressed, and after the sonication process, the purification process was monitored through SDS–PAGE in the supernatant (S), pellet (P), flowthrough (FT), wash (W), and elution (E) to finally obtain a high purity recombinant cldn18.2 (red arrow) (**d**).

**Figure 5 antibodies-13-00020-f005:**
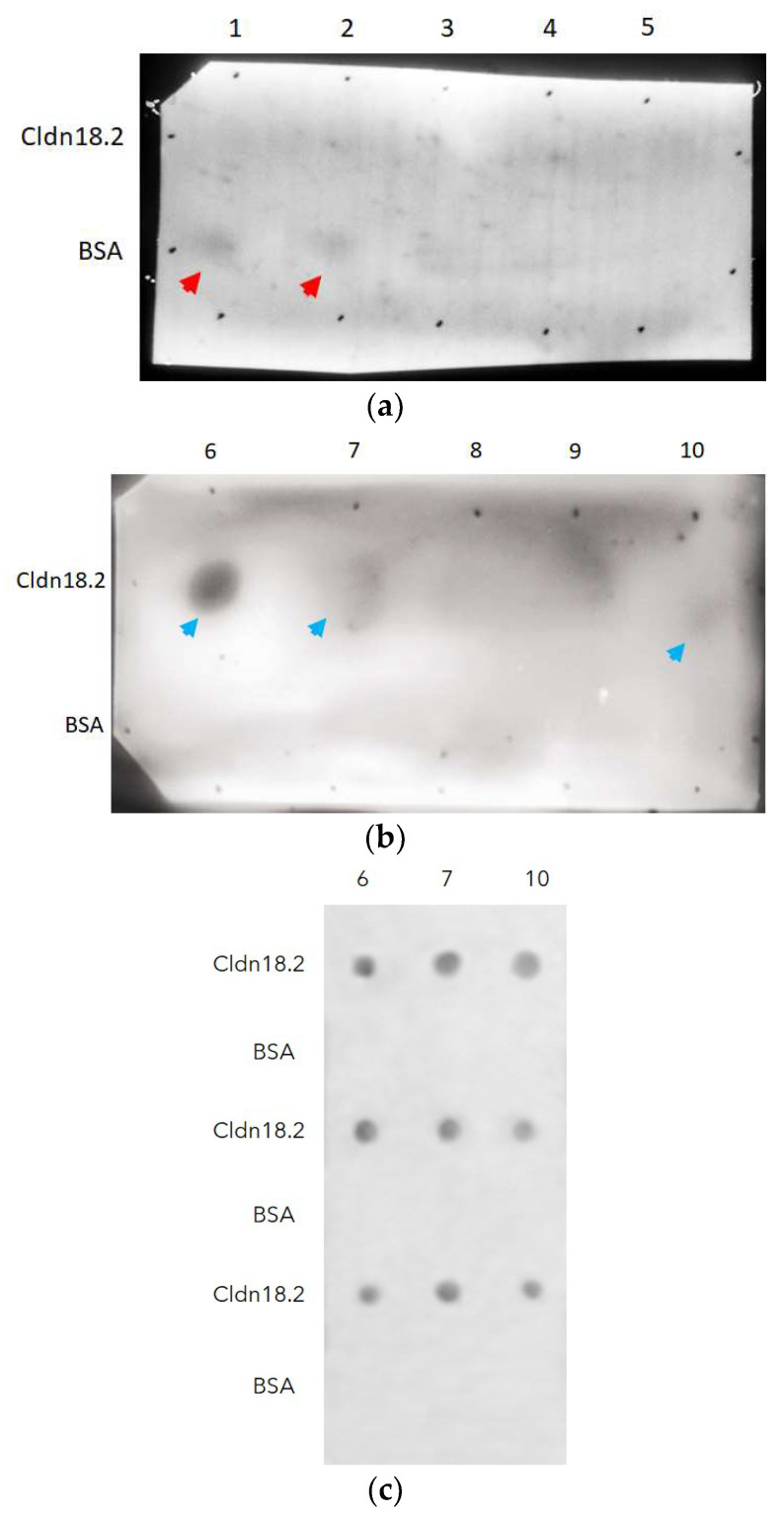
Dot-blotting analysis to identify from ten clones (lanes: 1–10) (**a**,**b**) specificity to cldn18.2 isolated in the third round of panning. Clones 1 and 2 (**a**) show slight specificity by BSA (head red arrows), but clones 6, 7, and 10 (head blue arrows) recognize cldn18.2 (**c**).

## Data Availability

The sequences obtained from *scFv* antibodies in this research are available from the corresponding author upon reasonable request.

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
