# Peer review of "Construction of a Human Immune Library from Gallbladder Cancer Patients for the Single-Chain Fragment Variable (scFv) Antibody Selection against Claudin 18.2 via Phage Display"

_2073-4468, 2024, doi:10.3390/antib13010020_

Round 1

Reviewer 1 Report

Comments and Suggestions for Authors

This investigation is merit. However, there are some suggestions and questions regarding the methodology.

  1. - ELISA was commonly used for clone-specific selection after three rounds of panning. A selection criterion with twice or more fold of ELISA OD value, compared to BSA, will be chosen. Why did the author choose to use Western blot, which is semi-qualitative, rather than ELISA?
  2. - How can we ensure that the isolated ScFv clone is specific to Claudin 18.2 without conducting a binding test?"

Author Response

Response letter Revisor 1

Reviewers´s comments:

This investigation is merit. However, there are some suggestions and questions regarding the methodology:

  • ELISA was communly used clone-specific selection after three rounds of panning. A selection criterion with twice or more fold of ELISA OD value, compared to BSA, will be chosen. Why did the author choose to use Western blot, which is semi-qualitative, rather than ELISA?

Answer: You are right. ELISA is the most commonly used method for this type of analysis. ELISA and western blot, or dot blot, are robust and sensitive methods, as they use antibodies for the specific detection of an antigen of interest; the difference, as you mention, is that in the ELISA you can quantify the reaction by means of DO readings; On the other hand, in the western blot or dot blot it is qualitative and only the color that the HRP develops with its substrate is observed. We did the western blot because we didn't have the TMB. We made the purchase request on January 16th from Thermo Fisher for the product 1-Step Ultra TMB ELISA. But the supplier made a mistake and sent us (on February 27th) the 1-Step Ultra TMB Blotting product (used for western blotting) (invoice attached), leaving us in trouble to be able to perform the ELISA analysis. We have already requested the product change, but that will take a significant amount of time.

Since we only kept the clones with an apparent positive signal (6, 7 and 10) (the others were discarded), we did dot blot only with this clones to confirm their reaction with claudin 18.2 immobilized in the PVDF membrane; resulting in figure 5c, which was added to the text on line 400 if you allow it as follow: (In the attached document)

How can we ensure that the isolated scFv clone is specific to claudin 18.2 without conducting a binding test?

Answer: Good point. We have the nucleotide sequence from all these three clones. We are working “in silico” analysis in order to create a tridimensional model of these three candidates and determine affinity by claudin 18.2 though molecular docking, and in this way edit the amino acids involved in the interaction with the claudin 18.2 and thus mature the affinity of the candidates. But this information will be considered in the next manuscript that we are preparing. In this paper we just want reported one functional immune library. 

Reviewer 2 Report

Comments and Suggestions for Authors

Dear authors.

I find your manuscript interesting, because gallbladder cancer is aggressive with no therapies and bad prognosis. Though early diagnosis is benefic. The manuscript targets the construction of a human immune library from PBMCs  of  patients with  gallbladder cancer. Also isolation of scFv-phage clones with specificity against the larger extracellular loop,  an important biomarker.

Tha introduction and discussion may be improved.

Methods and results are well structured.

Figure 5 must be reloaded or loaded improved.

Conclusions and references support the results and are fine.

Author Response

Response letter Revisor 2

Reviewers´s comments:

I find your manuscript interesting, because gallbladder cancer is aggressive with no therapies and bad prognosis. Though early diagnosis is benefic. The manuscript targets the construction of a human immune library from PBMCs of patients with gallbladder cancer. Also, isolation of scFv-phage clones with specificity against the larger extracellular loop, an important biomarker.

  • The introduction and discussion may be improved.

Answer: Thank you for this observation. The introduction has been reinforced talking about some mechanism of action of the mAb. The following paragraphs have been added between lines 55 – 62:

“The therapeutic efficacy of a mAb in cancer treatment varies depending on the tumor antigen and the diverse mechanisms of action of the mAb, ranging from receptor blockade to immune system activation and disruption of oncogenic signaling [9]. Mechanisms such as blocking ligand binding, conjugated mAbs, depending on the antibody conjugation, it can be either non-conjugated, blocking binding to evade undesired biological responses; or conjugated, harnessing the benefits provided by mAbs like vehicles and therapeutic agents such as cytotoxic molecules. In addition to mechanisms such as blocking signaling pathway and depletion of target by Fc Interaction [9].”

Lines 73 – 81:

“Phage display technology enables the expression and selection of high-affinity recombinant antibodies in laboratories in a controlled manner. These antibodies are fused with proteins from the phage coat and displayed on its surface, facilitating the isolation of scFv and Fab fragments targeting specific cellular markers [14]. Antibody libraries can be generated from pre-immune donors, using V domains from the germ line or synthetic genes [12,13]. The multivalent presentation of antibodies is crucial for efficient selection. Moreover, antibody selection in whole cells has proven effective in isolating antibodies targeting challenging antigens, such as those associated with tumors, multi-pass membrane proteins like cluster of differentiation or highly processed cell products [17]”.

Line 113 – 119:

“Under normal conditions in gastric tissue, the splice variant 2 of CLDN18 is located within tight junction complexes between gastric mucosal cells, and CLD18.2 epitopes have limited exposure to circulating antibodies. However, during the transition to a malignant state, loss of cellular polarity causes the CLDN 18.2 epitope to become exposed, making it more readily recognizable by antibodies [29]. The claudin-18.2 isoform represents a promising target for new experimental drugs such as zolbetuximab (IMAB362), currently under investigation in several clinical trials for advanced gastrointestinal tumors [29].”

  • Methods and results are well structured

Answer: Thank you for the observation

  • Figure 5 must be reloaded or loaded improved

Answer: We agree with your point. In fact, ELISA is the most commonly used method for this type of analysis. ELISA and western blot, or dot blot, are robust and sensitive methods, as they use antibodies for the specific detection of an antigen of interest; the difference, is that in the ELISA you can quantify the reaction by means of DO readings; On the other hand, in the western blot or dot blot it is qualitative and only the color that the HRP develops with its substrate is observed. We did the western blot because we didn't have the TMB. We made the purchase request on January 16th from Thermo Fisher for the product 1-Step Ultra TMB ELISA. But the supplier made a mistake and sent us (on February 27th) the 1-Step Ultra TMB Blotting product (used for western blotting) (invoice attached), leaving us in trouble to be able to perform the ELISA analysis. We have already requested the product change, but that will take a significant amount of time. (See the attached document)

Since we only kept the clones with an apparent positive signal (6, 7 and 10) (the others were discarded), we can´t do a new dotblot with the same clones; instead, we have tweaked the figure 5 and additionally we have incorporated a new dotblot with clones 6, 7 and 10 to confirm their reaction with claudin 18.2 immobilized in the PVDF membrane; resulting in figure 5c, which was added to the text on line 400 if you allow it as follow: (See the attached document)

(c)

  • Conclusions and references support the results and are fine

Answer: Thank you for the observation

Reviewer 3 Report

Comments and Suggestions for Authors

Author Response

Response letter Revisor 3

Reviewer´s comments:

The focus of the present work is the development of a human immune library from gallbladder cancer patients. The authors present the disadvantages of current approaches and successfully introduced phage display as a promising alternative. The background presented increases the importance of this work and the strategy followed is interesting; although not new.

Overall, the paper is well writer with all necessary information and details.

Some considerations that the reviewer has are:

  • Why did you choose PBMCs instead of other more specific population in the tumor microenvironment, for instance?

Answer: Thanks for your appreciation. The biological samples used in this research, were provided by the Biobanco belong to Pontificia Universidad Católica de Chile. Therefore, they separate the PBMCs from the other blood components and freeze it at -80°C. We don´t intervene in any step of this process. For this reason, we requested PBMCs, because it is also an important source of V genes to construct immune libraries, as widely described in the literature.  

  • Did you consider the clinical history of patients when selecting the samples?

Answer: We didn’t have access to the clinical history. The only condition we asked of the biobank was that they give us samples from patients with gallbladder cancer who have not started treatment, neither chemo nor radiotherapy; because this type of treatment could generate lymphopenia or low lymphocyte count.

  • Are you considering expressing some of these antibody fragments to assess the efficacy of the technique in identifying possible therapeutic molecules?

Answer: Yes, we are conducing “in silico” analysis in order to create a tridimensional model of these three candidates and determine affinity by claudin 18.2 though molecular docking, and in this way edit the amino acids involved in the interaction with the claudin 18.2 and thus mature the affinity of the candidates. But this information will be considered in the next manuscript that we are preparing. In this paper we just want reported one functional immune library.

  • An experimental validation is missing; which in the reviewer opinion is a major drawback

Answer: In this paper we just want reported one functional immune library. Validation experiment, characterization and another analysis are being conduced such as affinity maturation by point mutagenesis on CDRs regions, molecular docking, expression and purification, etc. 

Specific comments:

Title:

The reviewer is not sure if the authors can put “single format monoclonal antibody (scFv)” because it give us the idea the definition of scFv is single format monoclonal antibody when it is single-chain fragment variable.

Answer: Thanks for this suggestion. The title has been corrected as follows: “Construction of a human immune library from gallbladder cancer patients for single-chain fragment variable (scFv) antibody selection against claudin 18.2 via phage display”

Abstract:

It is well written and has all the necessary information to understand the scope of the work, the framework of the work, and the main results.

Line 25 – monoclonal antibodies (mAbs)

Answer: mAbs has been added on line 25

Introduction:

It has all the details and background.

Line 45 – specify the main therapeutic protocol

Answer: there aren´t therapeutic protocol to treat GBC. Neither chemotherapy nor radiotherapy respond well, because this kind of cancer is detected in advanced stages. For this reason, on line 46 we said: “… with surgical removal od the organ being the only preventive treatment”.

Line 51 – the mechanism described is for antibodies used specifically in GBC or in general? There are many different mechanisms of action of antibodies, such as, simple blocking, recruitment of immune system, direct toxicity. Authors should give more detail on this if they want to keep this paragraph.

Answer: We reforced this paragraph as follow on line 55 – 62: “The therapeutic efficacy of a mAb in cancer treatment varies depending on the tumor antigen and the diverse mechanisms of action of the mAb, ranging from receptor blockade to immune system activation and disruption of oncogenic signaling [9]. Mechanisms such as blocking ligand binding, conjugated mAbs, depending on the antibody conjugation, it can be either non-conjugated, blocking binding to evade undesired biological responses; or conjugated, harnessing the benefits provided by mAbs like vehicules and therapeutic agents such as cytotoxic molecules. In addition to mechanisms such as blocking signaling pathway and depletion of target by Fc Interaction [9]”.

Line 66 – and about the yield? Can you comment on that?

Answer: Phage display technology is just used to identify new antibodies sequences. Once the sequence identified, we must to use a recombinant expression platform according to the application that you want to the new antibody. So the production will depend that the recombinant platform that you have chosen. 

Line 86 – biomarket or biomarker?

Answer: Sorry for this mistake. Biomarket was changed by biomarker

Line 86 – describe what CBV means

Answer: Sorry for this mistake. We wanted say GBC. So, the word CBV was changed by GBC.

Line 86 – give more details on the importance of claudin 18 in healthy cells and cancer

Answer: Between lines 113 – 119 we added the following paragraph where we give more details about claundin 18:

Line 113 – 119: “Under normal conditions in gastric tissue, the splice variant 2 of CLDN18 is located within tight junction complexes between gastric mucosal cells, and CLD18.2 epitopes have limited exposure to circulating antibodies. However, during the transition to a malignant state, loss of cellular polarity causes the CLDN 18.2 epitope to become exposed, making it more readily recognizable by antibodies [29]. The claudin-18.2 isoform represents a promising target for new experimental drugs such as zolbetuximab (IMAB362), currently under investigation in several clinical trials for advanced gastrointestinal tumors [29]”.

Line 95 – this protein is already targeted by any drug or biologic? Is it only overexpressed in cancer cells? What is its role in normal cells?

Answer: There are one monoclonal antibody (Zolbetuximab - Chimeric) being evaluated to be used against claudin 18.2 in gastric or gastroesophageal junction adenocarcinoma. In order to explain about expression of the claudin in normal cells, the follow paragraph was added between lines 113 – 119:

Line 113 – 119: “Under normal conditions in gastric tissue, the splice variant 2 of CLDN18 is located within tight junction complexes between gastric mucosal cells, and CLD18.2 epitopes have limited exposure to circulating antibodies. However, during the transition to a malignant state, loss of cellular polarity causes the CLDN 18.2 epitope to become exposed, making it more readily recognizable by antibodies [29]. The claudin-18.2 isoform represents a promising target for new experimental drugs such as zolbetuximab (IMAB362), currently under investigation in several clinical trials for advanced gastrointestinal tumors [29]”.

Material and methods:

All the details are presented.

Line 106 – do you have access to the clinical history? Were these patients already treated with chemotherapy or other agent? Do you think that the treatment can affect the presence of PBMCs or at least the population that you need?

Answer: We didn’t have access to the clinical history. The only condition we asked of the biobank was that they give us samples from patients with gallbladder cancer who have not started treatment, neither chemo nor radiotherapy; because this type of treatment could generate lymphopenia or low lymphocyte count.

Line 198 – paragraph is too long. It is very difficult to read

Answer: the paragraph has been modified as follow:

Line 213 – 223: “A library aliquot, was mixed with protein loading buffer; then, it was denatured at 95°C for 5 min and loaded onto a 12% and 20% SDS – PAGE. Protein observation was performed by staining the gel with Coomasie brilliant blue. For the western blot analysis, proteins were transferred to a polyvinylidene fluoride (PVDF) membrane (Merck Millipore) in semi-dry electrotransfer system, using OwlTM HEP (Thermo Fischer Scientific) for 60 min at 20 volts. Western blot assay was done using Pierce® Fast Western Blot kit, ECL substrate (Thermo Fischer Scientific), following the manufacturer´s instructions. Primary and secondary antibody against his 6x (His-probe HIS.H8 sc-57598 and m-IgG2bBP-HRP – Santa Cruz Biotechnology) and simius virus 5 (SV5) (V5-Probe C-9 sc-271944 and m-IgG2aBP-HRP - Santa Cruz Biotechnology) tags (expressed in fusion with the scFv bound to the phages) were used.

Results:

The result are well presented and are easy to understand

Line 270 – what is a good integrity?

Answer: We meant good quality RNA, not degraded, observing the 28s and 18s subunits through an agarose gel. For clarity, on line 301, the word “integrity” was changed by “quality”.

Line 315 – information for the discussion or introduction

Answer: Thank you for this suggestion. The paragraph has been moved to discussion on lines 445 – 448 as follow:

Line 445 – 448: “In this sense, claudin 18.2 is highly expressed in tumors such as gastric cancer [35], pancreatic cancer [26], lung cancer [36], cholangiocarcinoma[37], and gallbladder cancer [38], and for this reason is one of the most clinically relevant target in cancer immunotherapy [39]”.

Discussion:

The discussion is well written and stands upon the solid results collected throughout the work

Line 358 – paragraph is too long

Answer: Thanks for this observation. We have structured this paragraph, adding punctuation marks, in order to generate shorter paragraphs as follow:

Line 404 – 424: “Immune libraries play a very important role in the selection of antibodies against specific disease antigens; and they are based on the immune repertoire collected through V genes from individuals infected or suffering from a particular disease [41].

However, this kind of libraries are biased to produce mAbs, usually of high affinity against antigens associated to an specific disease [42]; such as the constructed library reported here.

We used PBMCs from 7 patients with GBC to isolated V genes. To maximize complementarity, we used degeneracy primers designed by Kugler et al [31]; which can isolate different representative families of antibody V genes.

All these families were cloned into pCDisplay3 to ensemble the library. Our library size was compatible with the size reported to others constructed libraires, which range from 106 to 1010 [43–45]. However, it is very laborious and time consuming to build large libraries (> 1010), due to both the use of a large number of blood samples [46] and bacteria transformations [47,48].

Shui et al [44], built an immune library against liver cancer with blood samples from four patients, obtaining a library of size 1.7 x 107. In the case of Wu et al, they use just three lymph nodes samples from colorectal cancer patients to build an immune library. This final library was of 5.53 x 106.

In other hands, Dong et al [45], built a naïve library with more than four hundred healthy people, using for the library assembly a LoxP peptide as linker between the V genes to form scFv, in order to increase both the variability and the size of library, using cre-recombinase enzyme. The final size of the library made by Dong et al, was 1 x 1011.”

Round 2

Reviewer 3 Report

Comments and Suggestions for Authors

After reviewing the authors' answers, I think that everything is fine and ready for publication.